# Molecular Detection of *Phytophthora cinnamomi* by RPA-CRISPR/Cas12a-Mediated Isothermal Amplification

**Xiaoqiao Xu** [1], **Tingting Dai** [1,2,*], **Qin Xiong** [1], **Jing Yang** [2], **Jiahui Zang** [1] and **Tingli Liu** [3]

[1] South China Sustainable Forestry Co-Innovation Center, Nanjing Forestry University, Nanjing 210037, China; 8220110087@njfu.edu.cn (X.X.); xiongqin@njfu.edu.cn (Q.X.); zjh20011009@163.com (J.Z.)
[2] Advanced Analytical & Test Laboratory, Nanjing Forestry University, Nanjing 210037, China; yjnjfu@sina.com
[3] School of Food Science, Nanjing Xiaozhuang University, 3601 Hongjin Avenue, Nanjing 211171, China; liutingli@njxzc.edu.cn
* Correspondence: 13770647123@163.com

**Abstract:** Background: *Phytophthora cinnamomi* is one of the soil-borne pathogens that causes root rot and stem rot in many plants globally. *P. cinnamomi* has serious economic, social, and environmental impacts, threatening natural ecosystems and biodiversity. Methods: In this study, a molecular detection method based on Recombinant polymorphic amplification (RPA) combined using the CRISPR/Cas12a system was developed for *P. cinnamomi*. The method was found to be highly specific for *P. cinnamomi*. Results: The results showed that 10 *P. cinnamomi* isolates were positive; however, 21 *Phytophthora* species, 4 *Phytopythium* species, 18 fungal species, and 2 *Bursaphelenchus* species were negative. In total, 10 pg·$\mu$L$^{-1}$ of *P. cinnamomi* genomic DNA can be detected. The detection process is performed within 20 min at 37 °C, which makes it fast and convenient for use. Discussion: In conclusion, the RPA-CRISPR/Cas12a system in this study is a promising tool for the rapid and sensitive detection of *P. cinnamomi* in plant samples.

**Keywords:** *Phytophthora cinnamomi*; Rapid detection; Recombinase polymerase amplification; RPA-CRISPR/Cas12a

## 1. Introduction

*Phytophthora cinnamomi* is a soil-borne plant pathogen with a global distribution, and is listed as one of the 100 worst invasive alien species in the global invasive species database [1,2]. It infects nearly 5000 species of plants, including many plants important in agriculture, forestry, and horticulture [3]. *P. cinnamomi* is one of the most important pathogenic fungi in forest disease control, threatening natural ecosystems and biodiversity [4]. *P. cinnamomi* causes the decline of chestnuts and oaks in the Mediterranean region of Europe and the United States [5].

The first known species of *P. cinnamomi*, was described by Rands in 1922. *Phytophthora cinnamomi* var. *cinnamomi* (Rands) was found to cause stripe ulcer disease in camphor plants in Sumatra [6]. *P. cinnamomi* is widely distributed in America, Europe, Oceania, and Southeast Asia, and has been reported in more than 70 countries and regions. The host range is extremely broad, and 182 strains have been isolated from more than 3000 species of forest trees, agricultural crops, and other host plants. In China, the hosts of *P. cinnamomi* include pine needles, camphor, bromeliads, avocado, cedar, chili pepper, acacia, papaya, cinchona, tobacco, etc. [7].

The detection and analysis of *P. cinnamomi* has evolved with the development of molecular techniques, while its precision is being challenged by new agents [8]. Conventional PCR (real-time PCR) methods have been developed for the identification of *P. cinnamomi* using molecular detection tests based on isothermal amplification. Manisha B et al. (2019) developed a *P. cinnamomi* qPCR assay for detecting *P. cinnamomi*, which has high sensitivity [9]. However, these analytical methods rely on thermocycling devices and other

specialized equipment to prepare samples and provide results [10]. This also presents challenges, including primer interference, signal interference, and the complexity of system optimization. Furthermore, these traditional molecular assays have limited utility for disease diagnosis because they require centralized laboratory resources, expensive equipment, and complex sample preparation [11–14].

CRISPR-Cas (clusters of regularly spaced short palindromic repeats and their associated protein systems) technology has revolutionized the field of gene editing and has also shown great potential for various biological applications, including pathogen detection [15,16]. The CRISPR-Cas system is a powerful tool for detecting specific nucleic acid sequences with high sensitivity and specificity, making it an attractive option for the rapid and accurate diagnosis of diseases [17]. Researchers have developed CRISPR-Cas-based nucleic acid detection technology that is fast, accurate, and sensitive, making it ideal for next-generation rapid and sensitive on-site nucleic acid detection. The technology was recognized by Science as one of the top ten breakthroughs in science and technology in 2018 [18]. CRISPR-Cas has been successfully used for detecting pathogenic microorganisms, genetic diseases, tumor mutations, small molecules, etc. [19,20]. It was an ideal next-generation technology for the rapid and sensitive on-site detection of nucleic acids [21]. The use of CRISPR-Cas for pathogen detection offers several advantages over conventional methods such as PCR [22]. One of the main advantages is the speed of the assay. CRISPR-Cas systems can be designed to detect multiple targets simultaneously, reducing the time required for sample processing and analysis. Additionally, the system can be adapted for use in portable devices, enabling the on-site testing and real-time monitoring of infectious agents. Another advantage of CRISPR-Cas technology is its versatility [23–25]. The system can be programmed to target a wide range of pathogens. This makes it suitable for use in various settings, from clinical diagnostics to environmental monitoring [26]. Moreover, CRISPR-Cas technology has the potential to improve the accuracy of detection by reducing the chances of false positives and false negatives. The system can be designed to recognize specific sequences that are unique to a particular pathogen, minimizing the risk of cross-reactivity with other microorganisms [27,28]. In conclusion, CRISPR-Cas technology has shown great promise in the field of pathogen detection, offering advantages such as speed, versatility, and accuracy. As technology continues to evolve, it has the potential to transform the way we diagnose and manage infectious diseases, ultimately contributing to improved public health outcomes [29].

In this study, an RPA-CRISPR/Cas12a assay based on a novel target *Pcinn204169* using a comparative genomics approach was developed to detect *P. cinnamomi*, a devastating plant pathogen. This method was designed to provide fast results with a high accuracy, which is crucial for managing plant diseases effectively. Overall, this RPA-CRISPR/Cas12a assay offers a significant improvement in speed and convenience over traditional PCR-based methods for detecting *P. cinnamomi*. The ability to perform the entire process from sample to result in just 30 min makes it a valuable tool for rapid disease diagnosis and management, especially in forestry settings where timely intervention can prevent significant forest losses.

## 2. Materials and Methods

### 2.1. Maintenance of Isolates and DNA Extraction

The isolates of fungi and oomycetes tested in this study are listed in Table 1. The isolates were stored at the Department of Plant Pathology, Nanjing Forestry University (NFU), China. *P. cinnamomi* and other oomycetes were isolated and grew on a 10% concentration V8 vegetable juice agar on 70-mm Petri dishes at an optimum temperature of 18–25 °C. The fungi were isolated and grew on a potato dextrose agar (PDA) on 90-mm Petri dishes at an optimum temperature of 20 °C. Both were incubated in the dark. For 3–5 d, the genomic DNA (gDNA) of the test strain was extracted using the DNA secure Plant Kit (Tiangen Biotech, Beijing, China) and quantified using a Nanodrop ND-1000

spectrophotometer (NanoDrop Technologies, Wilmington, DE, USA). All DNA samples were stored in a refrigerator at −20 °C.

**Table 1.** Information and results of CRISPR-Cas12a detection in the *Phytophthora*, other oomycete isolates, and fungal isolates used in this investigation.

| Number | (Sub)Clade | Species | Isolate | Origin | | Crisp-Cas12a Deteciton Results |
|---|---|---|---|---|---|---|
| | | | | Host/Substrate | Source | |
| 1 | | *Phytophthora cinnamomi* | Pci1 | *Pinus* sp. | AH, China | + |
| 2 | | *P. cinnamomi* | Pci2 | *Rhododendron simsii* | JS, China | + |
| 3 | | *P. cinnamomi* | Pci3 | *Cedrus deodara* | JS, China | + |
| 4 | | *P. cinnamomi* | Pci4 | *Camellia oleifera* Abel. | JS, China | + |
| 5 | | *P. cinnamomi* | Pci5 | *Pinus* sp. | JS, China | + |
| 6 | | *P. cinnamomi* | Pci6 | *Rhododendron simsii* | AH, China | + |
| 7 | | *P. cinnamomi* | Pci7 | *Rhododendron simsii* | SD, China | + |
| 8 | | *P. cinnamomi* | Pci8 | *Cedrus deodara* | SD, China | + |
| 9 | | *P. cinnamomi* | Pci9 | *Cedrus deodara* | AH, China | + |
| 10 | | *P. cinnamomi* | Pci10 | *Pinus* sp. | SD, China | + |
| 11 | | *P. sojae* | Ps1 | *Glycine max* | JS, China | - |
| 12 | | *P. sojae* | Ps2 | *Glycine max* | JS, China | - |
| 13 | | *P. sojae* | Ps3 | *Glycine max* | JS, China | - |
| 14 | | *P. sojae* | Ps4 | *Glycine max* | JS, China | - |
| 15 | | *P. sojae* | Ps5 | *Glycine max* | JS, China | - |
| 16 | | *P. sojae* | Psf1 | *Glycine max* | FJ, China | - |
| 17 | | *P. sojae* | Psf2 | *Glycine max* | FJ, China | - |
| 18 | Oomycete | *P. sojae* | Psf3 | *Glycine max* | FJ, China | - |
| 19 | | *P. sojae* | Psf4 | *Glycine max* | FJ, China | - |
| 20 | | *p. medicaginis* | CBS 117685 | lucerne | Holland | - |
| 21 | | *p. medicaginis* | CBS 117689 | lucerne | Holland | - |
| 22 | | *P. parvispora* | CBS132771 | *Arbutus unedo* | Italy | - |
| 23 | | *P. parvispora* | CBS132772 | *Arbutus unedo* | Italy | - |
| 24 | | *P. cactorum* | C1 | *Malus pumila* | JS, China | - |
| 25 | | *P. cactorum* | C2 | *Malus pumila* | JS, China | - |
| 26 | | *P. cactorum* | C3 | *Rosa chinensis* | JS, China | - |
| 27 | | *P. nicotianae* | Pnl | *Nicotiana tabacum* | FJ, China | - |
| 28 | | *P. nicotianae* | Pn2 | *Lycopersicum* sp. | JS, China | - |
| 29 | | *P. pini* | Ppinil | *Rhododendron pulchrum* | JS, China | - |
| 30 | | *P. pini* | Ppini2 | *R. pulchrum* | JS, China | - |
| 31 | | *P. hibernalis* | CBS 270.31 | *Cirrus sinensis* | USA | - |
| 32 | | *Phytopythium litorale* | PC-dj1 | *Rhododendron simsii* | JS, China | - |
| 33 | | *P. helicoides* | PH-C | *Rhododendron simsii* | JS, China | - |
| 34 | | *P. helicoides* | PF-he2 | *Rhododendron simsii* | JS, China | - |
| 35 | | *P. helicoides* | PF-he3 | *Rhododendron simsii* | JS, China | - |

**Table 1.** *Cont.*

| Number | (Sub)Clade | Species | Isolate | Origin Host/Substrate | Source | Crisp-Cas12a Deteciton Results |
|--------|------------|---------|---------|------------------------|--------|-------------------------------|
| 36 | | *Fusarium. proliferatum* | Fprl | *Pinus* sp. | JS, China | - |
| 37 | | *F. oxysporum* | BMZ12185 | Cucumber Rootstocks | JS, China | - |
| 38 | | *F. lateritium* | BMZ51357 | *Freesia lanceolata* | JS, China | - |
| 39 | | *F. incarnatum* | IL3HQ | *Medicago sativa* | JS, China | - |
| 40 | | *F. acuminatum* | Facl | *Rhizophora apiculata* | SC, China | - |
| 41 | Fungi | *F. asiaticum* | Fasl | *Triticum aestivum* | JS, China | - |
| 42 | | *F. avenaceum* | Favl | *Glycine max* | JS, China | - |
| 43 | | *F. culmorum* | Fcul | *Glycine max* | SC, China | - |
| 44 | | *F. commune* | Fcol | Soil | HLJ, China | - |
| 45 | | *F. equiseti* | Feq1 | Soil | JS, China | - |
| 46 | | *F. lateritium* | Flatl | Soil | JS, China | - |
| 47 | | *F. moniforme* | Fmol | *Oryza sativa* | JS, China | - |
| 48 | | *F. redolens* | BMZ103188 | litter | JS, China | - |
| 49 | | *Colletotrichum gloeosporioides* | BMZ51334 | *Camellia japonica* | JS, China | - |
| 50 | | *Bursaphelenchus xylophilus* | Js-1 | *Pinus thunbergii* | JS, China | - |
| 51 | Nematode | *B. mucronatus* | Bmucro | *Pinus* sp. | JS, China | - |
| 52 | | *Endothia parasitica* | Epal | *Castanea mollissima* | JS, China | - |
| 53 | | *Bremia lactucae* | Blal | *Lactuca sativa* | JS, China | - |
| 54 | | *Gibberella avenacea* | BMZ105417 | oatmeal | JS, China | - |
| 55 | | *Gibberella tricincta* | BMZ102862 | rotten wood | JS, China | - |

*2.2. Designing RPA Primers, CRISPR RNA, and ssDNA Reporter Genes*

To select the candidate target genes for the *P. cinnamomi*-specific RPA–CRISPR reaction, the annotated genomic sequence of *P. cinnamomi* at https://genome.jgi.doe.gov/Phyci1/Phyci1.home.html (accessed on 7 March 2024) was retrieved. To identify the target genes unique to *P. cinnamomi*, we initially retrieved all the publicly available genome sequences of the *Phytophthoa* species. Then, all the 26,131 gene sequences of *P. cinnamomi* were used as the queries to blast against the above genomes (e-value cutoff: $1 \times 10^{-5}$), and the genes without any hit were treated as unique to *P. cinnamomi* [30]. The *Pcinn204169* gene (scaffold_174: 124989-125344) was selected as the target for the design of gene-specific RPA primers. The RPA primer was constructed according to the recommendation of the DNA sequencing kits manuals (Figure 1) using Primer Premier 6.0 (Premier Biosoft, Palm Alto, CA, USA). For the construction of the CRISPR RNA (crRNA) and ssDNA reporter, the CHOPCHOP web tool (http://chopchop.cbu.uib.no/, accessed on 26 May 2023) was employed. The crRNA was designed not to duplicate the sequence of the RPA primers and targeted the conservative region in the RPA amplification (crRNA: UAAUUUCUACUAAGUGUA-GAUAGGCCAAUGCCGCCAAUGAC) (Figure 1). The 5′ terminus of the ssDNA reporter was labeled using 6-FAM, whereas the 3′ terminus was labeled using the BHQ-1 transcription factor quencher (5′-6-FAM-TTATT-BHQ-1-3′). The crRNA and ssDNA reporters were prepared by GenScript (Nanjing, China). They were maintained at −80 °C until the assay was carried out.

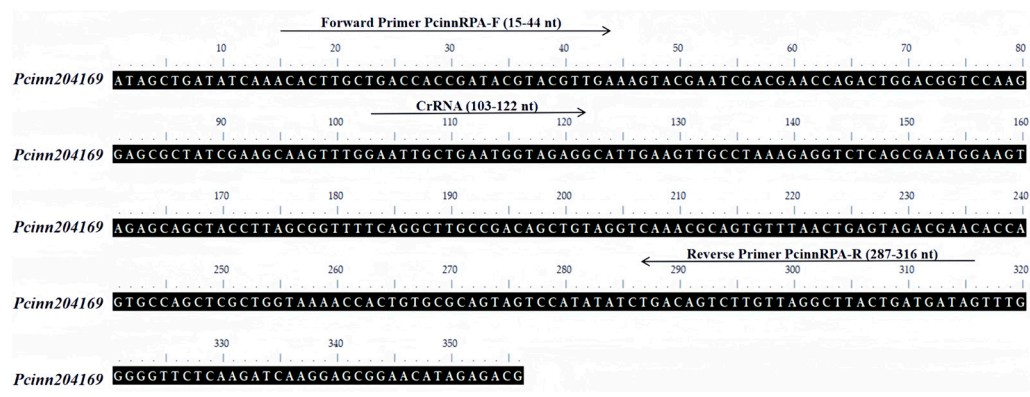

**Figure 1.** Sequence analysis of *Pcinn204169* from *Phytophthora cinnamomi*. Nucleotides targeted by the forward (*Pcinn204169* RPA-F) and reverse (*Pcinn204169* RPA-R) primers; the crRNA sequences are shown below the respective arrows. The arrows indicate the direction of amplification.

### 2.3. RPA–CRISPR/Cas12a Assays

The 20 min test consisted of two steps: 10 min for the RPA reaction and 10 min for the detection of CRISPR/Cas12a. Firstly, *P. cinnamomi* DNA was amplified in 10 min using a pair of RPA primers (*Pcinn204169* RPA-F/-R). Then, the amplified product was detected and visualized within 10 min using the CRISPR/Cas12a system. The flow of the assay for the analysis of RPA-CRISPR/Cas12a is illustrated in Figure 2. The assays were conducted according to the manufacturer's recommendations using the Test Strip Kit (LeShang Ltd., WuXi, China) in a 50 μL reaction mixture. Initially, the reactions consisted of 2 μL of each forward or backward primer (*Pcinn204169* RPA-F/-R, 10 μM), 25 μL of the supplied buffer, 2 μL of gDNA (100 ng/μL), and 16 μL of doubly distilled water (ddH$_2$O) (47 μL total). Each sample was centrifuged at 4000× *g* for 5 s, and then 3 μL of initiator was added to the cap of the reaction tube. The tube was covered and centrifuged at 4000 rpm for 5 s after multiple rounds of mixing, and the reaction was performed at 37 °C for 20 min. We spent 20 min on the detection test. To eliminate false positives, each set of reactions included a no-template control (NTC) and a positive control (PTC). The CRISPR/Cas12a system was used to analyze RPA products. RPA amplification products were used as templates and various concentrations of crRNA (40, 300, and 500 nM and 1, 2, 5, as well as 10 μM) and ssDNA (40 and 300 nM and 2, 1, 5, as well as 10 μM) were tested for optimum combinations of concentrations. The optimization of the reaction time of the RPA treatment and cleavage of Cas12a was tested at eight different times (5, 10, 15, 20, 25, 30, 35 and 40 min). The CRISPR/Cas12a expression was carried out using a 50 μL volume containing 38 μL ddH$_2$O, 5 μL reaction buffers, 3 μL crRNA, 1 μL of Cas12a, 1 μL of ssDNA, and 2 μL of RPA products. The tube was then vortexed for 5 s at 4000 rpm and the reaction was performed at 37 °C. There are two methods for detecting the outcome of the RPA-CRISPR/Cas12a test. The multifunctional microplate reader detects a strong fluorescent signal or a visibly emitted green, fluorescent light under a transmitted blue LED light at about 470 nm, while no fluorescent signals or visibly emitted green, fluorescent light are observed for the negative controls. At least three replicates were performed for all RPA-CRISPR/Cas12a reactions. The STDEV function was used to calculate the standard deviation for the three CRISPR/Cas12a analysis results (1, 2, and 3). Statistical analysis was performed using GraphPad Prism 8 software (GraphPad software Inc., San Diego, CA, USA). For the analysis of variance, the experimental group was analyzed against the control group and a *p*-value was calculated. The difference of $p < 0.05$ (*) was considered statistically significant.

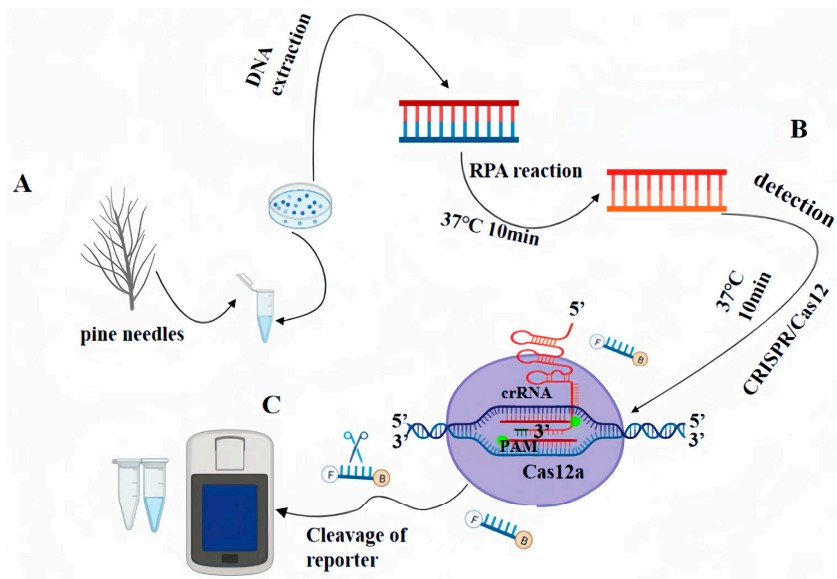

**Figure 2.** Schematic diagram of the RPA-CRISPR/Cas12a assay used for the detection of *Phytophthora cinnamomi*. (**A**) Recombinase polymerase amplification (RPA). (**B**) Cas12a protein can bind to any amplicon and target-specific CRISPR RNA (crRNA) to form a complex. The complex has indiscriminate ssDNA cleavage activity. The FAM-labeled ssDNA reporter is cleaved. When excited at 470 nm, the reporter emits visible green fluorescence. (**C**) Positive result: green fluorescence is visible. Negative result: green fluorescence is not visible.

### 2.4. RPA–CRISPR/Cas12a Assay Specificity and Sensitivity

To evaluate the specificity of the RPA-CRISPR/Cas12a assay for *P. cinnamomi*, closely related *Phytophthora* species, such as *P. parvispora*, *P. hibernalis*, *P. sojae*, *P. pini*, *P. cactorum*, and *P. nicotianae*, were tested to evaluate the specificity. Intergeneric species such as *P. ittorale*, *P. helicoides*, *P. aphanidermatum*, *P. dissotocum*, *F. oxysporium*, and *F. solani* were also tested to further validate the accuracy of the specificity. Each set of reactions consisted of purified gDNA samples (100 ng) as the template, positive control (100 ng *P. cinnamomi* isolate), and ddH$_2$O as NTC. We employed seven serial dilutions of *P. cinnamomi* genomic DNA (e.g., 10 ng, 1 ng, 100 pg, 10 pg, 1 pg, 100 fg, and 10 fg) as templates for both the conventional PCR and RPA-CRISPR/Cas12a assays to evaluate the sensitivity of this assay. In both techniques, we incorporated an NTC into each reaction set and executed three replicates for all template concentrations. The STDEV function was used to analyze and calculate standard deviation. GraphPad Prism 8 software was used for statistical analyses. An analysis of variance was performed by comparing the experimental group with the control group and calculating the *p*-values. The difference of $p < 0.05$ (*) was considered statistically significant.

### 2.5. RPA-CRISPR/Cas12a to Detect P. cinnamomi from Artificially Inoculated Pine Needles

On the campus of Nanjing Forestry University, healthy pine needle plants were cut, rinsed with water for approximately 20 min, and placed on a piece of sterilized filter paper for the surface sterilization of pine needle blades. Pine needle stalks were inoculated with *P. cinnamomi*, incubated wet, and observed daily. On day 1, there was no significant change in pine needle stems. On day 2, pine needle stems were slightly blackened, and on days 3 and 4, the discoloration of the plant stems was more pronounced. The diseased pine needle stalks were cut into small pieces of approximately one cubic centimeter. Specimens were ground for 1 min using a grinder. To extract DNA, 1 mL of sodium hydroxide lysate and 100 mg of inoculated sample were mixed vigorously in liquid nitrogen powder for 10 min at room temperature (25 °C). During incubation, the samples were tapped three times. RPA (100 ng·μL$^{-1}$) and ddH$_2$O were used as PTC and NTC, respectively, and 2 μL of lysate was used as a template for the RPA assay. At least three repetitions of the experiment

were performed. The STDEV function was used for the analysis and calculation of the standard deviation. GraphPad Prism 8 software was used for statistical analyses. The experimental group was analyzed against the control group and a *p*-value was calculated for the analysis of variance. The difference was $p < 0.05$ (*), which is statistically significant.

## 3. Results

### 3.1. Optimization of the RPA-CRISPR/Cas12a-Based Test for the Detection of P. cinnamomi

Overall, 10 μM crRNA and 10 μM ssDNA reporters were optimized for the expression of RPA-CRISPR/Cas12a, as shown by the intensity of green fluorescence and fluorescence (Figure 3A,B). The optimal concentrations were tested to determine the optimal RPA reaction time (assessed at 5, 10, 15, 20, 25, 30, 35, and 40 min). After 10 min, a considerable amount of green fluorescent emission could be detected under the blue light LED illuminator (Figure 4A,B). The optimal response time for RPA was 10 min. The optimal cleavage time (5, 10, 15, 20, 25, 30, 35, and 40 min) of Cas12a was evaluated using the 10 min RPA reaction product, and the results showed that the optimum clearing time was at 10 min (Figure 4C,D). In summary, the green fluorescent protein is visible after only 20 min (10 min for the RPA and 10 min for the Cas12a cleavage) (Figure 4).

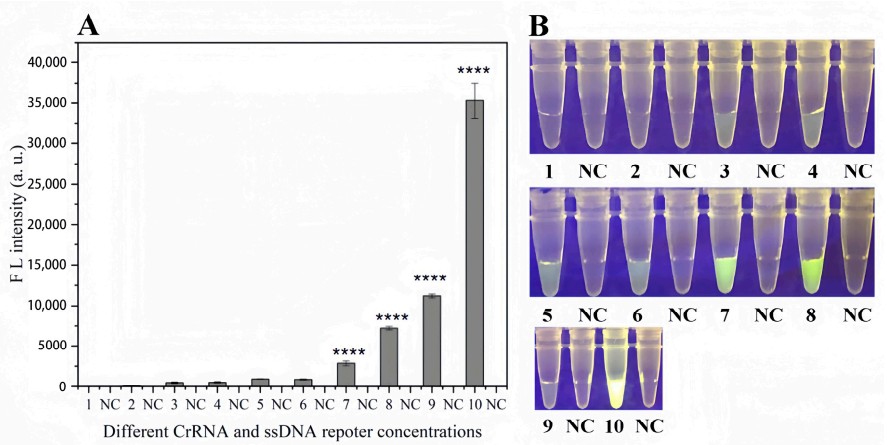

**Figure 3.** Optimization of the RPA-CRISPR/Cas12a assay for the detection of *Phytophthora cinnamomi*. The crRNA and ssDNA reporter concentrations were set to the following values: 1: 40 nM crRNA, 40 nM ssDNA reporter; 2: 300 nM crRNA, 40 nM ssDNA reporter; 3: 500 nM crRNA, 300 nM ssDNA reporter; 4: 1 μM crRNA, 1 μM ssDNA reporter; 5: 1 μM crRNA, 2 μM ssDNA reporter; 6: 2 μM crRNA, 1 μM ssDNA reporter; 7: 5 μM crRNA, 1 μM ssDNA reporter; 8: 1 μM crRNA, 5 μM ssDNA reporter; 9: 10 μM crRNA, 5 μM ssDNA reporter; and 10: 10 μM crRNA, 10 μM ssDNA reporter. NC, negative control (double-distilled $H_2O$). (**A**) Fluorescence detection using a multifunctional microplate reader (λex: 485 nm, λem: 520 nm). (**B**) Visible green fluorescence detection under a blue LED transilluminator at 470 nm. The one-way ANOVA of the fluorescence readings with those of the negative control showed that $p < 0.0001$ (****). "****" shows a significant difference between fluorescent and non-fluorescent signals.

### 3.2. The Specificity of P. cinnamomi Was Rapidly Detected with RPA-CRISPR/Cas12a

*P. cinnamomi* was used as the template DNA, with primers *Pcinn204169* RPA-F/-R. Interspecific DNAs are shown from left to right: *P. parvispora*, *P. hibernalis*, *P. sojae*, *P. pini*, *P. cactorum*, and *P. nicotianae* (Figure 5A,C). Intergeneric DNAs are shown from left to right: *P. litorale*, *P. helicoides*, *P. aphanidermatum*, *P. dissotocum*, *F. oxysporium*, *F. solani*, and the blank control NC (Figure 5B,D). When testing the specificity of the RPA–CRISPR/Cas12a test, the multi-functional microtiter plate readers demonstrated a high fluorescent response to the *P. cinnamomi* gDNA, while the gDNA from the other oomycetes and fungi did not fluoresce.

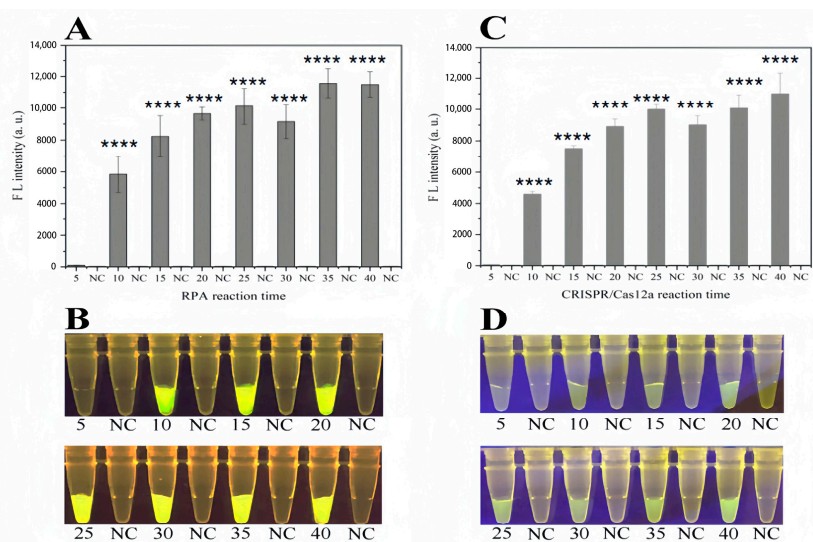

**Figure 4.** Optimization of the recombinase polymerase amplification (RPA) reaction time and Cas12a cleavage time for the RPA–CRISPR/Cas12a assay. (**A**,**B**) RPA reaction times: 5, 10, 15, 20, 25, 30, 35, and 40 min; NC (negative control, ddH$_2$O). (**C**,**D**) Cas12a cleavage times: 5, 10, 15, 20, 25, 30, 35, and 40 min; NC (negative control, ddH$_2$O). (**A**,**C**) Fluorescence detection using a multifunctional microplate reader (λex: 485 nm, λem: 520 nm). (**B**,**D**) Detection of green fluorescence in the visible range under a blue LED transmitted light illuminator with a wavelength of 470 nm. The one-way ANOVA of the fluorescence readings with those of the negative control showed that $p < 0.0001$ (****). "****" shows a significant difference between fluorescent and non-fluorescent signals.

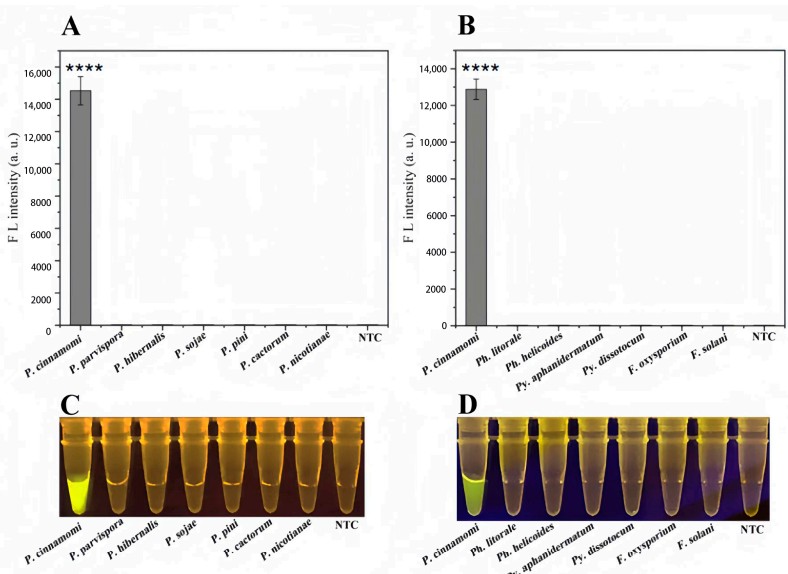

**Figure 5.** The specificity of *Phytophthora cinnamomi* was rapidly detected with the RPA-CRISPR/Cas12a assay. (**A**,**B**) Evaluation using genomic DNA isolated from *P. cinnamomi*, *P. parvispora*, *P. hibernalis*, *P. sojae*, *P. pini*, *P. cactorum*, *P. nicotianae*, and negative control (NC; no template). (**C**,**D**) Evaluation using genomic DNA from *P. cinnamomi*, *P. litorale*, *P. helicoides*, *P. aphanidermatum*, *P. dissotocum*, *F. oxysporium*, *F. solani*, and the negative control (NC; no template). The one-way ANOVA of the fluorescence readings with those of the negative control showed that $p < 0.0001$ (****). "****" shows a significant difference between fluorescent and non-fluorescent signals.

### 3.3. The Sensitivity of P. cinnamomi Was Rapidly Detected with the RPA-CRISPR/Cas12a Assay

To determine the sensitivity of the RPA-CRISPR/Cas12a rapid assay for detecting *P. cinnamomi*, different concentrations of *P. cinnamomi* gDNA were sequentially diluted, and

different concentrations of *P. cinnamomi* gDNA (10, and 1 ng; 100, 10, and 1 pg; and 100 fg) were extracted for the experiments. Concentrations of gDNA ranging between 10 ng, 1 ng, 100 pg, and 10 pg showed a visible green, fluorescent emission, while other levels of gDNA and the NC showed no visual evidence of fluorescence (Figure 6). Consistent values were obtained in three replications. This indicates that 10 pg is the minimum detectable concentration of gDNA.

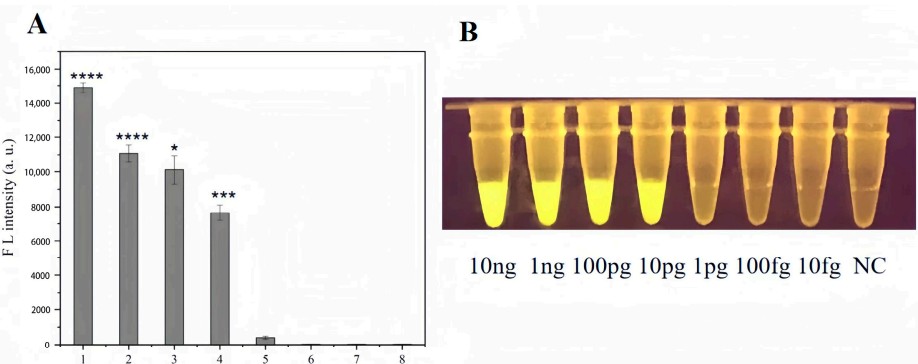

**Figure 6.** The sensitivity of *Phytophthora cinnamomi* was rapidly detected with the RPA-CRISPR/Cas12a assay. The pathogen was detected at a minimum genomic DNA concentration of 10 pg using (**A**) a multifunctional microplate reader (λex: 485 nm, λem: 520 nm) or (**B**) a blue LED transilluminator at 470 nm. 1–8: 10 ng, 1 ng, 100 pg, 10 pg, 1 pg, 100 fg, 10 fg, and NC (negative control, double-distilled H$_2$O). The one-way ANOVA of the fluorescence readings with those of the negative control showed that $p < 0.0001$ (****); $0.001 < p < 0.01$ (***); $p < 0.05$ (*), "****"; "***"; "*" shows a significant difference between fluorescent and non-fluorescent signals.

### 3.4. Experimental Detection of P. cinnamomi-Infested Pine Needle Plants Using the RPA-CRISPR/Cas12a Approach

The crude DNA from inoculated pine needle stems of 1, 2, and 3 d and non-inoculated pine needle stems was extracted by NaOH lyes. For the RPA-CRISPR/Cas12a assay, the extracted DNA was used as a template. Purified gDNA (100 ng/µL) was used as the positive control and dd H$_2$O was used as the negative control. *P. cinnamomi* was detected, as confirmed by the green fluorescence of the RPA-CRISPR/Cas12a rapid assay, in crude DNA samples from the positive controls and plants inoculated with *P. cinnamomi* (Figure 7A–C). Thus, RPA–CRISPR/Cas12a can effectively detect artificially inoculated *P. cinnamomi.*

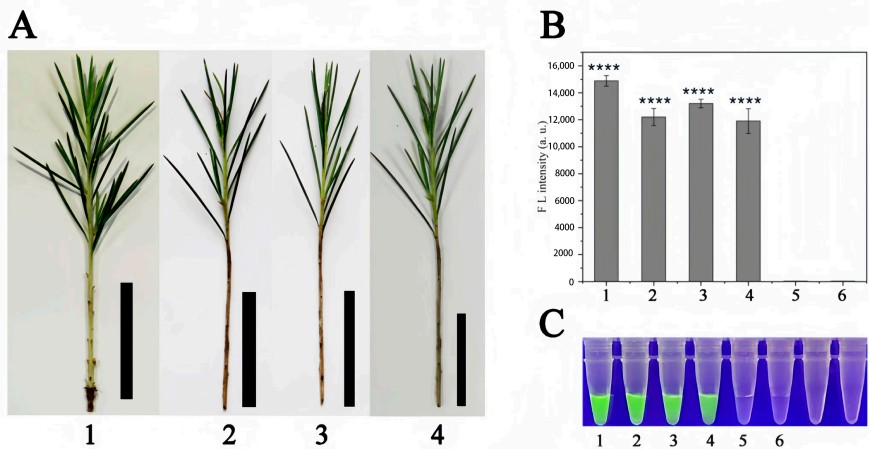

**Figure 7.** Experimental detection of *Phytophthora cinnamomi*-infested pine needle plants using the RPA-CRISPR/Cas12a approach. (**A**) Artificially inoculated pine needles (1: healthy pine needle stems; 2–4: Samples of naturally infected pine needle stems). (**B**) Strong fluorescence signals were detected

by a multifunctional microplate reader ($\lambda$ex: 485 nm, $\lambda$em: 520 nm). 1: Positive control (*P. cinnamomi* gDNA), PTC; 2–4: samples of naturally infected pine needle stems; 5: healthy pine needle stems; NC: negative control. (**C**) Visible green fluorescence was detected under a blue LED transilluminator at 470 nm. 1: Positive control (*P. cinnamomi* gDNA); 2–4: samples of naturally infected pine needle stems; 5: healthy pine needle stems; NC: negative control. The one-way ANOVA of the fluorescence readings with those of the negative control showed that $p < 0.0001$ [****]. "****" shows a significant difference between fluorescent and non-fluorescent signals.

## 4. Discussion

The RPA-CRISPR/Cas12a rapid diagnostic test for *P. cinnamomi* is reported here for the first time. The overall reaction times were 20 min (10 min for the RPA reactions and 10 min for the CRISPR/Cas12a assay), and the assay results could be visualized using UV lighting and fluorescence reading. The RPA-CRISPR/Cas12a assay was strongly influenced by the concentration of crRNA and ssDNA. Therefore, we screened various combinations of crRNA and ssDNA reporter gene concentrations, with both at a concentration of 10 $\mu$M. Our specificity evaluation confirms that the novel RPA-CRISPR/Cas12a system accurately detected *P. cinnamomi* gDNA and was negative for gDNA from other species. The method specifically detects *P. cinnamomi* at a minimum gDNA content of 10 pg using an RPA-CRISPR/Cas12a assay. The pine needle samples collected from *P. cinnamomi*-infected needles were used to evaluate the feasibility of the assay.

To date, for the detection of *P. cinnamomi* by nucleic acids, the characteristics of these methods are summarized as follows: qPCR is highly sensitive for the detection of *P. cinnamomi* but requires thermal cycling conditions and temperatures above 95 °C [9]. The detection of *P. cinnamomi* by qPCR can be performed in the field by using a portable qPCR instrument, which can be used in the field. Although portable qPCR instruments can be used in the field, the high temperatures and thermal cycling temperatures provided by high-power batteries or automobile batteries affect their feasibility in field testing [29,30]. LAMP is an efficient isothermal nucleic acid technique, but the need for high-power batteries affects its application in field testing due to the need for reaction temperatures in the 55 to 60 °C range [31].

The RPA-CRISPR/Cas12a method has many advantages over traditional techniques. On the one hand, the reaction is able to be completed in a stable temperature range at a fairly constant level of 37 °C. Such conditions can be easily achieved using human body temperature, a USB-powered incubator, or a thermostatic heater, this eliminates the need to use specialized and expensive instruments like temperature cycling devices. On the other hand, it is possible to perform the diagnostic procedure in ~20 min, whereas PCR-based tests require at least 2.5 h, with the PCR procedure taking 90 min and the gel electrophoresis 30 min [32–35]. And third, some RPA reaction components interfere with the antibodies on the test paper, which can lead to an inappropriate dilution, non-specific bindings, and incorrectly diluted false-positive reactions. The RPA binding with CRISPR-Cas12a allows for the dual detection of the target: once for identifying the RPA primer and then for identifying the RPA amplification product by CRISPR/Cas12a during the RPA reaction. This effectively circumvents the problem of false positives during RPA amplification [36,37]. RPA-CRISPR/Cas12a detection systems have many advantageous features, such as multiplexing capabilities, low requirement for complicated instrumentation and specialized personnel, and superior detection accuracy [38–42]. In contrast to the RPA-LFD approach, RPA-CRISPR/Cas12a allows for the isothermal, label-free target gene detection through the design of RPA primer sets and amplification of the products as amplified reporter genes [43–45]. The resulting CRISPR/Cas12a cleavage products provide a precise, cost-effective, and easy-to-use platform for the development of instant CRISPR-based diagnostic applications [46,47].

The specific primers were designed to amplify the *Pcinn204169* putative gene, a new target identified from genomic sequence data. This *Pcinn204169* RPA-CRISPR/Cas12a assay has been demonstrated to have specificity for Pcin. It also has several other advantages

compared to PCR-based detection assays. The assay is convenient and transportable, without the need for costly equipment. However, this study did not examine continuous base changes or deletions, leaving considerable opportunity to investigate the ability of RPA-CRISPR/Cas12a to distinguish individual differences in bases. The CRISPR/Cas12a assay developed here can be used in the laboratory to detect and characterize *P. cinnamomi*, and the accuracy of this assay was verified using pine needle samples collected at Nanjing Forestry University. In addition, RPA-CRISPR/Cas12a rapidly detected *P. cinnamomi* in <30 min at 37 °C, greatly reducing the detection time.

## 5. Conclusions

In this study, a CRISPR/Cas12a-mediated isothermal amplification technique was established to detect *P. cinnamomi* that did not cross-react with other interspecies and intergeneric DNAs during specificity experiments and had good specificity. The method established in this study is highly sensitive and capable of detecting 10 pg of the *P. cinnamomi* genome DNA in 20 min of incubation to 37 °C. The method is sensitive, efficient, and convenient, and the results are clearly visible under UV light, which is favorable for the early detection of *P. cinnamomi.*

**Author Contributions:** X.X.: Investigation, Data curation, Formal analysis, Writing—review and editing; Q.X.: Formal analysis; J.Y.: Data curation; J.Z.: Formal analysis; T.D.: Supervision, Project administration, Writing; T.L.: Funding acquisition, review and editing. All authors have read and agreed to the published version of the manuscript.

**Funding:** This work had support from China national key R&D program (2023YFD1401304), natural sciences funding from Jiangsu province (BK20231291), Jiangsu University Natural Science Research Major Project (21KJA220003), and the Priority Academic Program Development of Jiangsu Higher Education Institutions.

**Data Availability Statement:** All data from this investigation are the subject of this article.

**Conflicts of Interest:** The authors declare no conflict of interest.

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
