# Peer review of "Molecular Detection of Phytophthora cinnamomi by RPA-CRISPR/Cas12a-Mediated Isothermal Amplification"

_forests, doi:10.3390/f15050772_

Round 1

Reviewer 1 Report

Comments and Suggestions for Authors

This manuscript is interesting they present the first RPA CRISPR-CAS12a isothermal assay to Phytophthora cinnamomi. The tools is interesting and will be a good new tool to have for detection of this important pathogens. However I think many information, references, addition to M&M is need to improve the manuscript.

Specific comments:

First recheck all reference #, I think many are not correct or not in right order wtih the right refences ex) L40 ref 7 I think probably should be ref 6?Same for reference 10 that are not host but Covid reference?...Ref 12 wrong and many others.

Also check typos few in the manuscript and English, ex)L58 "Th hosts" should be "the"; L47 change "ovipositor" for "oospore" the ovipositor are for insects...

L62 Should add more info on qPCR for P. cinnamomi available, and then compare the sensitivity of your assays to them.

Also recently quite few RPA CRISPR-Cas12 assays have been developed for Phytopthora species, P. ramorum, P. sojae and P. cambivora... Should cite and mention it and compare it in discussion, does your sensitivity that I think not very high comparable to it. Seems you have 10pg/ul and whith qPCR we can do usually around 200fg that is way more sensitive? Should have a comparison assays with other method in parallele I beleive.

    • 10.3389/fcimb.2023.1208837  ;  

https://doi.org/10.3389/fcimb.2023.1218105  ;

     https://doi.org/10.3390/f14112141

    L114 change "fruit" for "vegetable"

    Table 1, please reorganized column 2 (Sub clade) as it is not well assigned or aligned with the right Species group and lines.

    L124-135, This section is lacking a lot of information, still at the end I dont know what gene or genic regions was used for this assay, ITS? ribosomal DNA or other, single copy or multicopy genes? Any alignment we can  have with other species to see differences and SNPs? Accession #? Very not clear and need info.

    All figure are blury but of in supplement image material.

    L183 and other in the manuscript does the "." in "ng.ul-1 " is the way the journal is using this? Also all dilution info need to be add in this section 2.4 we see it only in results later?, how many dilutions...

    Also you have inuculated Pines but not clear is real field samples with P. cinnamomi were used some confusion with one figure and section, should see if your sensitivity can detect real samples?

    Figure 3 legend and others the way the time is presented can be confusing?

    L257 This info on Standard curve genomics should be in M&M not there? 

    L287, Figure 6 A, I dont follow the # here 1 to4 what they mean does not seems to fits with the 1 to 6 in B and C??? confusing. Also what is the positive control 1? and 2-4 naturally infected pine needles, it is infectec trees or from inuculation?

    L297 what you mention Cedrus deodara you made the experiment with Pinus, where it come from?  Also do you know what 10pg is in number of spore maybe. Comparison other qPCR and CRISPR-CAS12 RPA assays or LAMP? Region use should other region or multiplex prepare?

    L363 "Video"?

    Recheck all reference citation in text# does not seem to match.

    Maybe Figure S1 will be important in manuscript.

    Author Response

    Please refer to the attached document.

    Reviewer 2 Report

    Comments and Suggestions for Authors

    Please improve the abstract and revise the writing style: Please add this info in the abstract: Introduction, Problem and Objective of the study (missing in abstract), Method, Results, and conclusion. 

    Please remove Lines 25-31/32, which are not coherent with the Phytophthora cinnamomi story, good to have the introduction of Phytophthora and detection techniques. 

    Refortmat Table 1 to the scientific table the current table is difficult to read.

    Please use high-resolution figures for all. The present figures are not clear and sharp. 

    Please remove lines 297 -306 not related to the discussion of the study, This paragraph is more suitable in the introduction. 

    Lines 340-341 should be in the introduction not in discussion. Please rephrase if you would like to keep it here. 

    Comments on the Quality of English Language

    Please ask the author to improve the writing as in my comment. 

    Author Response

    Please refer to the attached document.

    Round 2

    Reviewer 1 Report

    Comments and Suggestions for Authors

    The authors have responded in their response to all comments however compare to the response, the changes in the manuscripts should be more developed and details. They are very short and do not satisfy to the request. Again should recheck spelling was supposed to have answer my comments on Line 37 " In China, Th hosts of P. cinnamomi include pine needle..." "Th" should be "the". Also when ask to get more info on qPCR need more than one reference and get more info on it, also in discussion about qPCR L306-314 do not reference anything and can be a better comparison explanation. The information on the gene used is better "Pcinn204169 gene was selected" however how this gene is ID from what genome, also any info on potential homology with an other gene name for it. Seem coming like this from nowhere and difficult to associate it, need more info. I think more work still need in the manuscript.

    Author Response

    Please refer to the attached document.
